# Development of Uniform Polydimethylsiloxane Arrays through Inkjet Printing

**DOI:** 10.3390/polym15020462

**Published:** 2023-01-16

**Authors:** Ning Tu, Jeffery C. C. Lo, S. W. Ricky Lee

**Affiliations:** 1Department of Mechanical and Aerospace Engineering, The Hong Kong University of Science and Technology, Hong Kong SAR, China; 2Foshan Research Institute for Smart Manufacturing, The Hong Kong University of Science and Technology, Hong Kong SAR, China; 3Electronic Packaging Laboratory, The Hong Kong University of Science and Technology, Hong Kong SAR, China; 4Smart Manufacturing Thrust, The Hong Kong University of Science and Technology, Guangzhou 511458, China; 5HKUST Shenzhen-Hong Kong Collaborative Innovation Research Institute, Shenzhen 518045, China; 6HKUST LED-FPD Technology R&D Centre, Foshan 528200, China

**Keywords:** inkjet printing, polydimethylsiloxane, quantum dots, additive manufacturing, color conversion

## Abstract

The inkjet printing method is a promising method to deposit polymer and functional nanoparticles at the microscale. It can be applied in the fabrication of multicolor polymer light emitting diodes (polyLEDs), polymer base electronics, multicolor color conversion layers, and quantum dot light emitting diodes (QLEDs). One of the main challenges is to print high-resolution polymer dots from dilute polymer solution. In addition, the quality of printed multicolor polyLEDs, QLEDs and multicolor color conversion layers is currently limited by non-uniformity of the printed dots. In this paper, polydimethylsiloxane (PDMS) is selected as the functional polymer, due to its high transparency, good reflective index value, inflammable and flexible properties. The optimal ink to form a uniform PDMS dot array is presented in this paper. Both the solvent and PDMS were tuned to form the uniform PDMS dot array. The uniform PDMS dot array was printed with a diameter of around 50 µm, and the array of closely spaced green quantum dots (QDs) mixed with PDMS ink was also printed on the substrate uniformly. While the green QD-PDMS film was printed at a resolution of 1693 dpi, the uniformity was evaluated using the photoluminescence (PL) spectrum and color coordinate value.

## 1. Introduction

Inkjet printing is a common method to transfer functional materials to paper or transparent film. In addition, it is commonly used in the office or household. In recent years, many efforts have been devoted to making inkjet printing a versatile method in additive manufacturing processes [1,2,3,4]. The inkjet printer can deposit a predetermined amount of materials in the desired position, which makes inkjet printing a highly potential manufacturing method for the fabrication of functional electronic devices [5,6,7]. Meanwhile, polydimethylsiloxane (PDMS) is a widely used in biological technology and optical technology, due to its optical transparency, biocompatibility, relatively high reflective index properties, and low cost. However, new applications are bringing new challenges. Inkjet printing requires low-viscosity inks, thus PDMS can only be printed through dilute solution. However, the drying droplet always faces the coffee-ring effect problem, which is caused by the combination of the pinning of the contact line and the higher evaporation rate at the edge of the contact line. The solutes are driven to the edge of the contact line due to capillary flow [8]. The ring-like PDMS dots affect the printing quality of the printing device. Thus, improving the uniformity of PDMS dots is of great practical interest. Mikkonen et al. presented an inkjet printable PDMS solution for patterning multilayered electronic structure. The PDMS was printed on the surface of the silver lines, in order to demonstrate the printable multilayer electrical devices [9]. However, uniform PDMS dots were not achieved. This is because inkjet printing is very vulnerable to blockages due to solvent drying. Hence, a printing method that relies on rapid evaporation of solvent is intrinsically difficult to maintain [10].

In this paper, we demonstrate the PDMS polymer inks based on a mixture of toluene and decane. We systematically varied the mixing ratio of the two solvents with PDMS in order to create a uniform drying droplet. The uniform PDMS dot array is also demonstrated, with an average diameter of around 50 µm with a standard deviation of 2.1. In addition, the PDMS polymer ink can also be mixed with quantum dots to form QD-PDMS ink. A green QD-PDMS dot array and film are also deposited on the substrate uniformly. The green QD-PDMS film was printed at a resolution of 1693 dpi, which makes high-resolution polymer inkjet printing a viable strategy for flexible electronics and intelligent packaging.

## 2. Materials and Methods

To obtain a uniform PDMS polymer dot array, the viscosity, surface tension, vapor pressure, and density of materials present a unique challenge for ink preparation. With the development of wearable electronics and flexible electronics, inkjet printing of PDMS on rigid substrate is of more interest than on paper [9,10,11]. The first challenge to prepare PDMS ink is to find a compatible solvent, that PDMS can work with. The compatibility of PDMS solvent can be considered in three aspects: the first is the swelling properties of PDMS in solvent, the second is the segmentation of solutes between PDMS and solvent, and the third is PDMS oligomers dissolved in a solvent. Jessamine et al. indicated that the swelling of PDMS has the greatest influence [12]. PDMS swells well in the solvent once the polymer ink is prepared. One of the parameters for predicting the solubility of PDMS is cohesive energy density, *c* (cal/cm^3^), which can be expressed as:(1)$$c=-\frac{U}{V}$$

*U* is the molar internal energy (cal/mol), and *V* is the molar volume (cm^3^/mol). If two materials are soluble, their cohesive energy density would be similar. For cross-linked polymers such as PDMS, that do not dissolve, solubility is measured by the degree of swelling, since this energy must be overcome to separate the molecules of the solute to allow the molecules of the solvent to insert polymer [13,14,15,16]. The cohesive energy density can also be expressed in terms of Hildebrand value, *δ* (cal^1/2^/cm^3/2^), for predicting the swelling properties of a polymer in a solvent. The Hildebrand value can be expressed as:(2)$$\delta =c^{1/2}={\left(-\frac{U}{V}\right)}^{1/2}$$

Hexane, chloroform, chlorobenzene, toluene, and decane were selected as the solvent to prepare the PDMS ink. Figure 1a is the swelling relationship of these solvents with PDMS. The dashed line is the Hildebrand value of PDMS at room temperature (*δ* = 7.3 cal^1/2^/cm^3/2^). For a solvent with a Hildebrand value closer to 7.3 cal^1/2^/cm^3/2^, the PDMS will have a greater swelling degree. Thus, PDMS exhibited the greatest swelling degree in hexane, but the worst swelling degree in both chlorobenzene and chloroform. Besides the solubility of PDMS, printability also needs to be considered. Considering hexane will easily block the printing nozzle due to its easy volatility at room temperature, toluene and decane were selected as two potential solvents to prepare the polymer ink. Figure 1b is the surface tension vs. vapor pressure of selected solvents at room temperature. Toluene is more volatile than decane at room temperature. This indicates that toluene has a higher potential to block nozzles than decane. Thus, decane was selected as the solvent to dissolve PDMS.

The PDMS was purchased from SYLGARO 184 from Dow Corning. The toluene (ACS reagent, ≥99.5%) was purchased from Sigma-Aldrich (St. Louis, MO, USA), while the decane (ACS reagent, ≥99.5%) was purchased from Sigma-Aldrich as well. The Green CdS/ZnS QDs were purchased from Mesolight Inc. (Suzhou, China). In addition, the inkjet printer applied was a FUJI Dimatix DMP-2850. The printing droplet volume was fixed as 10 pl, while the printing height was 0.5 mm with a droplet printing velocity of 1.9 m/s.

Sapphire was applied as the printed substrate. The sapphire was cleaned in detergent, DI water, acetone, DI water, and IPA in sequence, each for 10 min under ultrasonication. The cleaned sapphire was dried before use in the experiment. The ink viscosity and ink surface tension were studied to obtain a stable printing result. The viscosity measurement spin speed was 200 rpm with an accuracy of ±0.03 cP. The ink contact angle was measured by a Biolin Scientific Optical Tensiometer at room temperature, while the ink viscosity was measured by a Brookfield viscometer DV2T at room temperature.

## 3. Results and Discussion

Figure 2a shows the deposit of PDMS ink with decane as the solvent on the sapphire substrate. After drying, a regular PDMS dot is formed, while Figure 2b shows the cross-section of the PDMS dots in the x-direction and y-direction. Figure 2b shows that the width of the dot is 30.8 µm while the height is around 0.12 µm. The polymer dot has less polymer in the center of the dot than at the edge of the polymer dot. The polymer tends to move to the edge of the dots, caused by capillary flow. Once the droplet is deposited on the substrate, the solvent starts to evaporate, while the evaporation flux along the deposit droplet surface is not uniformly distributed [5]. Figure 3 is the schematic image of the drying process.

The edge of the droplet has a higher evaporation rate than the center of the droplet, as there is higher diffusion freedom from the center of the droplet to the edge of the droplet due to the different evaporation rate of the droplet [17]. Next, the capillary flow is formed from the center of the droplet to the edge of the droplet, indicated by the black arrows in Figure 3a. Thus, the PDMS dot with only decane solvent tends to form a crater shape as shown in Figure 2. In consideration of the capillary flow in the single droplet drying process, a co-solvent was chosen to reduce the high evaporation rate from the center to the edge by introducing the Marangoni effect [18]. The Marangoni effect phenomenon is known to affect how liquid spreads on rigid surface. The Marangoni effect (also called the Gibbs-Marangoni effect) is the surface tension gradient at the interface of two liquids. How the Marangoni effect influences the liquid spreading process was studied by Dora Pesach et al., who found that the spreading behavior of a mixture of solvents depends on the spreading behavior of the individual solvents [19]. The Marangoni flow can induce a droplet to spread in the capillary flow direction or slow a droplet spreading in the capillary flow direction, by just changing the physical properties of the two-solvent mixture [20]. If *P_a_* < *P_b_* (*P_a_* is the vapor pressure of solvent *a*, *P_b_* is the vapor pressure of solvent *b*), and *γ_a_* < *γ_b_* (*γ_a_* is the surface tension of solvent *a*, *γ_b_* is the surface tension of solvent *b*), solvent b tends to increase the surface tension at the edge of the droplet, causing the *Marangoni flow* to be in the opposite direction to the capillary flow [21,22,23].

In order to evaluate this formulaic assumption, toluene was selected as the co-solvent to dissolve PDMS with decane at room temperature. Toluene has higher vapor pressure and larger surface tension values than decane at ambient conditions, as shown in Figure 1b. Furthermore, both the Hildebrand value of toluene and decane are closer to 7.3 cal^1/2^/cm^3/2^, as shown in Figure 1a. The mixing ratio of the two solvents is systematically studied in this paper. The concentration of the PDMS was fixed at 10 mg/mL, while the mixing volume ratio of toluene and decane was changed. Figure 4 shows the morphology of PDMS dots printed on clean sapphire with different decane volume ratios. Through the 3D morphological image in Figure 4A1–A3, it can be observed that the dots’ morphology changed with the changing of the volume ratio of decane. When the volume ratio of decane was 20%, the PDMS dots exhibited a significantly coffee-ring effect, indicating that most of the PDMS gathered at the edge of the dots. Under this condition, the *Marangoni flow* is not sufficient to balance the capillary flow. After increasing the volume ratio of decane to 50%, the PDMS dots disperses on the substrate uniformly, without exhibiting the coffee-ring effect. Once the droplet was deposited on the substrate, toluene at the edge evaporates first due to its higher vapor pressure value. The rest of the solvent approached a higher concentration of lower vapor pressure solvent as the droplet drying process proceeded, which lowered the diffusion freedom from center to edge. At the same time, the surface tension gradient also formed a Marangoni flow from edge to center. As the toluene at the edge will evaporate first, the surface tension tends to decrease as decane has a lower surface tension than toluene during the drying process, which leads to a surface tension gradient from edge to center (Figure 3b). During the drying process, decane gradually becomes the dominant solvent in the remaining droplet, so the surface tension gradient no longer persists. Uniform dots were achieved. However, once the concentration of decane increased to 60%, the PDMS dots tended to take on unstable shapes, and the PDMS tended to gather at the edge of dots. In addition, the center of dots exhibited irregular depressions, as shown in Figure 4A3. Increasing the volume ratio of decane will also increase the effect of Marangoni flow, which will upset the balance between Marangoni flow and capillary flow. The diameter of decane with a 60% volume ratio is 41.7 µm. The diameters of PDMS dots with 20% decane and 50% decane were around 50 µm, larger than 41.7 µm, which is due to the change of the contact line affected by the Marangoni flow from edge to center.

Besides the effects of volume ratio, the effects of different PDMS concentrations were also evaluated. Figure 5 is the PDMS dots printed on sapphire substrate with different PDMS concentrations. The mixed solvent was fixed with toluene combined with decane at a volume ratio of 1/1 *v/v*. The concentration of PDMS affects the polymer dots’ morphology as well as mixed solvent [24]. When the concentration was only 5 mg/mL, the morphology of the PDMS dots was dispersed on the printed substrate discretely. There was a small dot in the center with several irregular dots surrounding the center. With further increases in the concentration of the PDMS to 10 mg/mL, the morphology of the PDMS dots dispersed more evenly on the rigid substrate. Once the concentration of PDMS was increased to 20 mg/mL, the morphology of the PDMS printed dots separated into two or three small dots surrounded by several even smaller dots. Subsequently once the PDMS concentration was increased to 30 mg/mL, the PDMS dots tended to form a single dot in the center with several small dots surrounding it. However, the position of the center dots was not always in the center of the print, as some center dots moved to an edge position. The increasing concentration of PDMS affected the contact angle of the ink, which was then affected again by the evaporation flux at the edge of the print contact line. Figure 6a illustrates the ink viscosity and ink contact angle with different PDMS concentrations. The contact angle increased with the increasing of PDMS. A smaller contact angle tended to have higher evaporation flux, and better wettability compared with a larger contact angle. Thus, the contact line of the dots changed when the PDMS concentration changed. Additionally, the PDMS swelling ability was also affected when changing the PDMS concentration. The ink viscosity increased with the increasing of PDMS concentration. This is because the viscosity of the solvent mixture is only 0.8 g/mL, much smaller than the viscosity of PDMS. Thus, increasing PDMS results in increasing the intrinsic viscosity. The diameter of PDMS dots exhibited a downward trend with the increasing of PDMS concentration as illustrated in Figure 6b.

The defined edge of the pinned area is the dots’ diameter (*D*), while the width of the dots is the width of dominant dots (*w*). The ratio (*r*) is the width of dominant dots divided by the diameter of the dots:(3)$$r=\frac{w}{D}$$
when the width was equal to the diameter, the PDMS droplet dispersed uniformly on the substrate. In this way, the more closely the ratio approaches to 1, the more consistently the PDMS dots will be distributed.

After this systematic evaluation of the effects of solvents and polymers, it was possible to experimentally achieve a uniform PDMS printed dot array. Figure 7 shows the diameter distribution of PDMS dots. We take 100 dots to count the average diameter of PDMS dots, and the average diameter value was 50.6 µm with a standard deviation (SD) value of 2.1. Meanwhile, green QDs were added into the PDMS ink at a concentration of 25 mg/mL [25]. Figure 8 shows the diameter distribution of the green QD-PDMS dot array under UV excitation. According to the image, the green QDs disperse on the substrate uniformly without the coffee-ring effect problem. The average diameter of printed green QDs was around 49.2 µm with a SD value of 1.9 which was less than approximately 5% of the average diameter value. In the meantime, green QD-PDMS film was printed at a resolution of 1693 dpi. The green QD-PDMS film emitted green color uniformly. The optical properties of green QD-PDMS film are show in Figure 9. The green QD-PDMS films were measured at five positions, which were upper left corner, lower left corner, upper right corner, lower right corner, and center of the film. The PL spectrums were found to be almost the same even with different measurement positions. The wavelength was kept at 541 nm with the FWHM value of 25 nm. The intensity of the spectrum had little variation in different positions which may be due to the irregular distribution of the backlight. The color coordinate value of green QD-PDMS film at different positions is shown in Figure 9B2. We can see that the color coordinate value in the *x*-axis direction is within a range of 0.204 and 0.205 with only 0.001 difference in x value, while the color coordinate value in the *y*-axis direction was in the range of 0.7505 and 0.7525 with only a 0.002 difference in the *y*-axis direction. The color coordinate value was similar to the QD-PDMS film with different measurement positions, and PL spectrum overlapping with little difference in color intensity. Overall, the printed green QD-PDMS film was uniform.

## 4. Conclusions

A well-defined high-resolution PDMS dot array was achieved by studying PDMS ink design. PDMS ink based on a mixture of decane, and toluene can improve the inkjet printing quality of PDMS. The coffee-ring effect was reduced by the addition of toluene as a co-solvent and the introduction of Marangoni flow. The mixing ratio of decane to toluene was systematically varied to obtain uniform polymer dots. In addition, the concentration of PDMS was varied to achieve a stable and uniform polymer dot array as well. PDMS ink can also work as a base ink for mixing with QDs, and inkjet-printed high-resolution PDMS QD arrays and uniform QD-PDMS films. Inkjet printing of high-quality QD-PDMS films makes printing high-quality color conversion films a promising method in the display packaging area. As a result, inkjet printing of high-resolution PDMS dot arrays could be a practical method for the fabrication of high-resolution wearable electronics in a fast and convenient way.

## Figures and Tables

**Figure 1 polymers-15-00462-f001:**
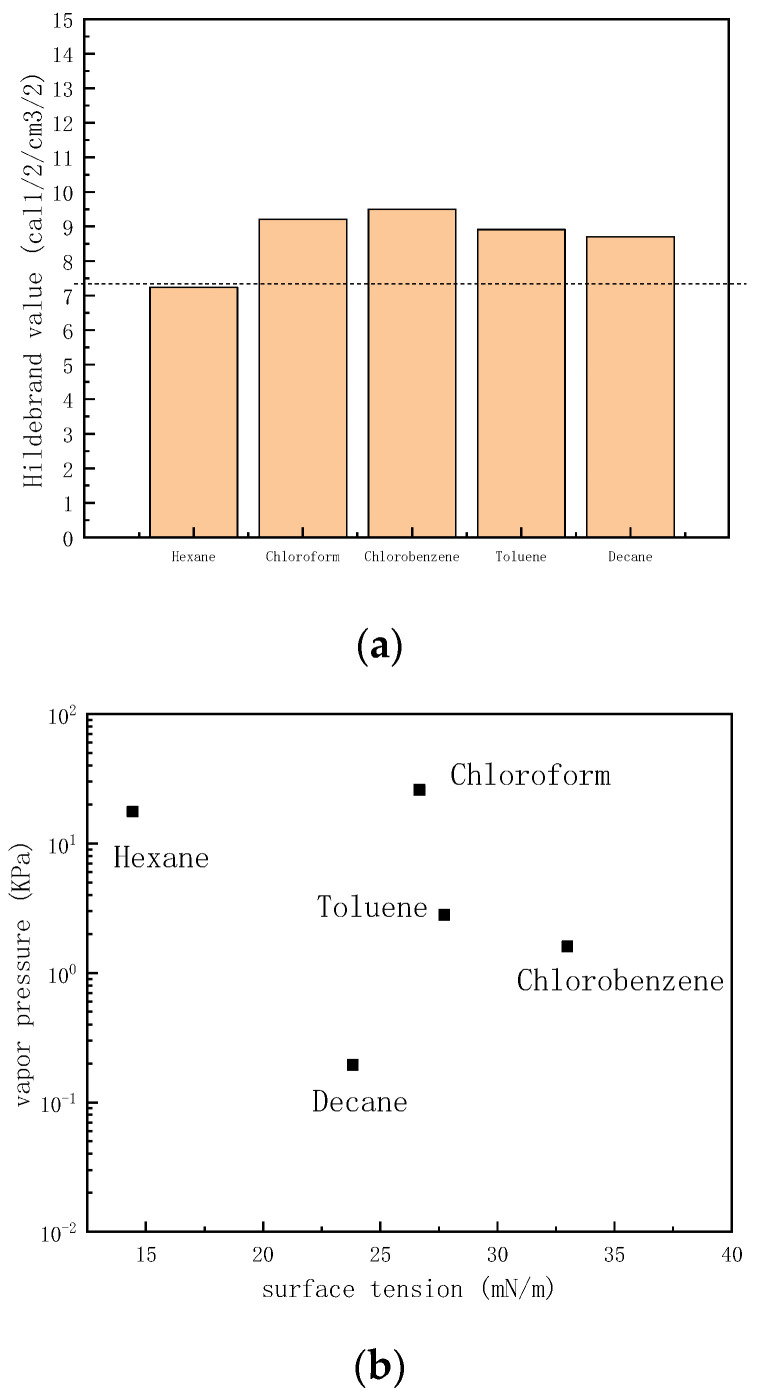
(**a**) Swelling relationship between PDMS and several solvents (The dashed line is the Hildebrand value of PDMS, *δ* = 7.3 cal^1/2^/cm^3/2^), (**b**) Surface tension (*γ*) vs. vapor pressure (*P_v_*) of selected solvent at room temperature.

**Figure 2 polymers-15-00462-f002:**
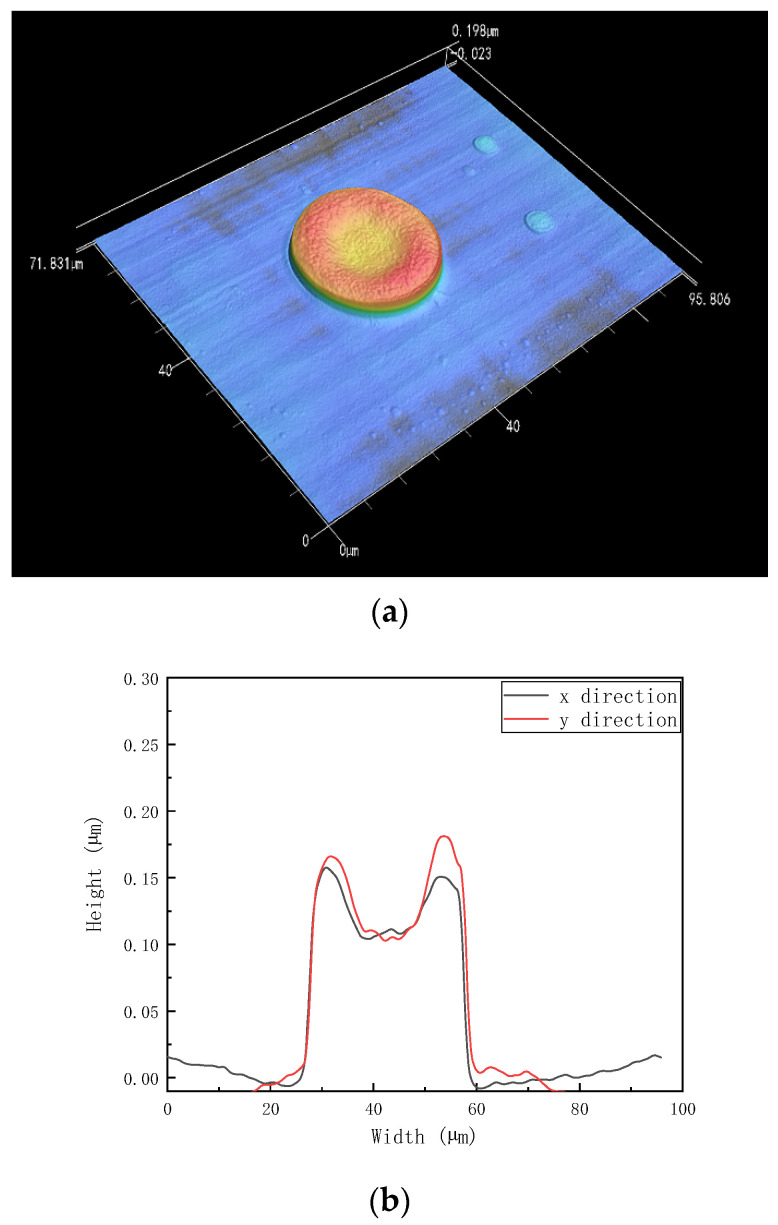
(**a**) PDMS polymer dot, formed by an inkjet print droplet of 10 mg/mL PDMS in decane on sapphire; (**b**) Cross-sections of the PDMS dots in the x-direction and y-direction.

**Figure 3 polymers-15-00462-f003:**
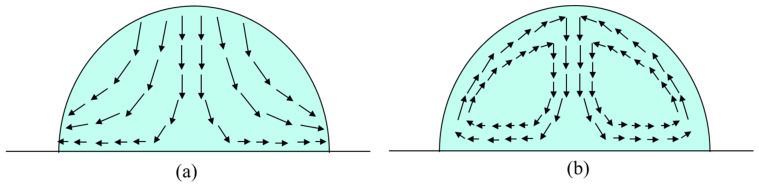
The drying process of a PDMS droplet. (**a**) Fluid flux motion of single-solvent PDMS ink under ambient conditions, (**b**) Fluid flux motion of PDMS ink combined with a co-solvent under ambient conditions.

**Figure 4 polymers-15-00462-f004:**
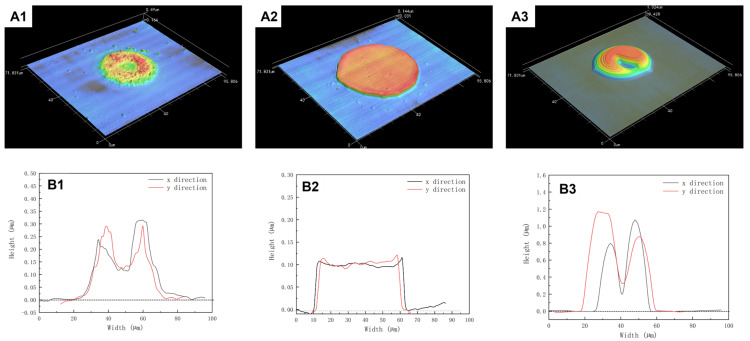
PDMS dots printed on sapphire substrate with different volume ratios. (**A1**–**A3**) 3D morphology image of PDMS dots, (**B1**–**B3**) PDMS dots’ cross section profile in the x-direction and y-direction. The volume ratio of decane from (**A1**–**A3**) is 20%, 50%, and 60%.

**Figure 5 polymers-15-00462-f005:**
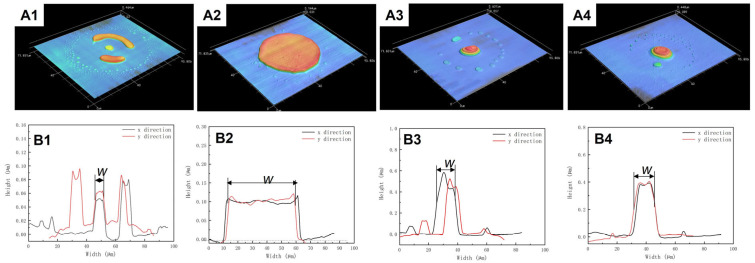
PDMS dots printed on a sapphire substrate with different concentrations of PDMS. (**A1**–**A4**) 3D morphological image of a single PDMS dot, (**B1**–**B4**) PDMS dots’ cross-section profile in the x-direction and y-direction. The concentration of the PDMS from (**A1**–**A4**) was 5 mg/mL, 10 mg/mL, 20 mg/mL, and 30 mg/mL in sequence.

**Figure 6 polymers-15-00462-f006:**
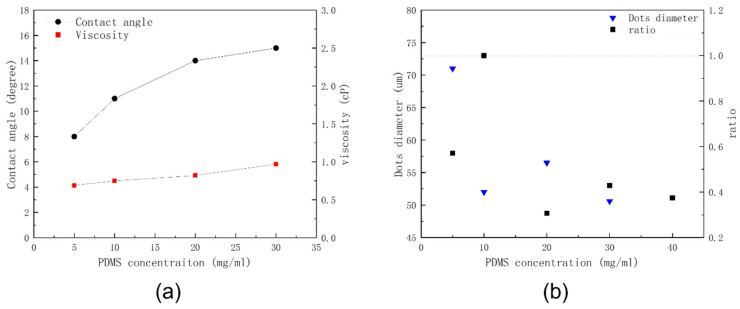
(**a**) Ink viscosity and ink contact angle with different PDMS concentrations at room temperature (25 °C), (**b**) PDMS dots’ diameter and ratio of PDMS width to PDMS diameter with different PDMS concentrations.

**Figure 7 polymers-15-00462-f007:**
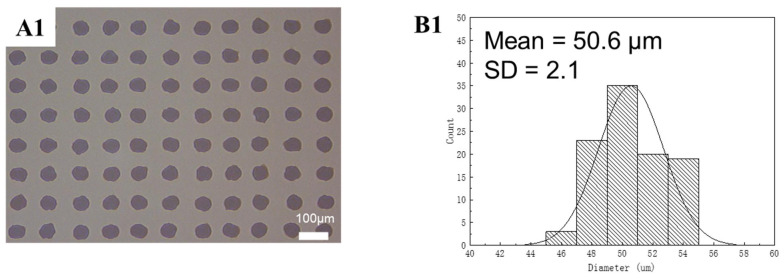
(**A1**) The microscope image of the printed PDMS dot array on sapphire, (**B1**) Diameter distribution of the printed PDMS dot array (from arrays of 10 × 10) on sapphire. The standard deviation (SD) was less than 2.53 or approximately 5% of the mean.

**Figure 8 polymers-15-00462-f008:**
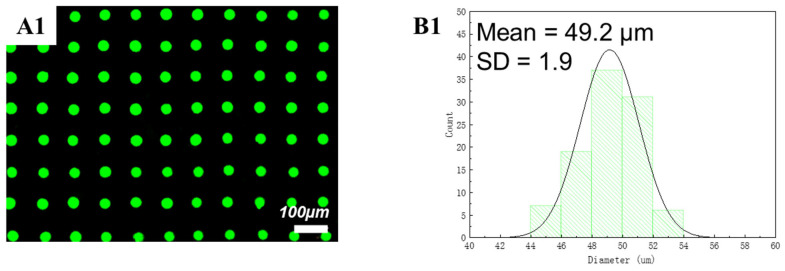
(**A1**) The microscope image of the printed green QD-PDMS dot array on sapphire under 365 nm UV excitation. (**B1**) Diameter distribution of the printed green QD-PDMS dot array (from array of 10 × 10) on sapphire. The standard deviation (SD) was less than 2.46 or approximately 5% of the mean.

**Figure 9 polymers-15-00462-f009:**
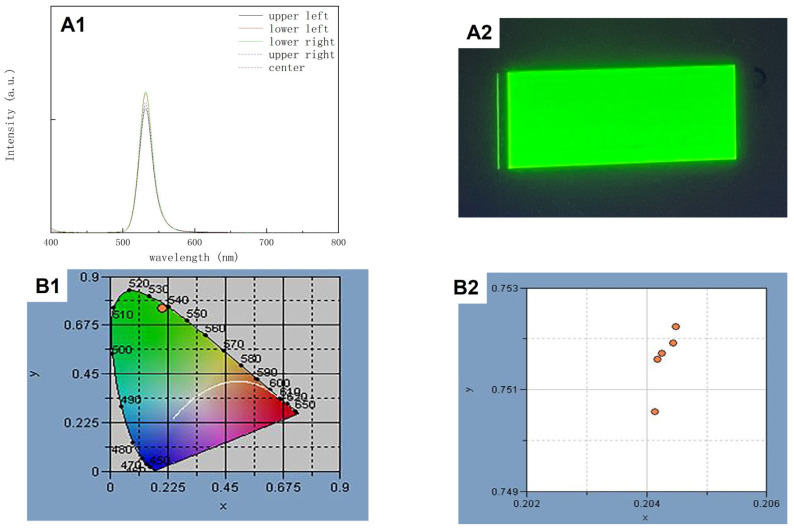
(**A1**) The photoluminescence (PL) spectrum of the printed green QD-PDMS film at different positions. (**A2**) Microscope image of the printed green QD-PDMS film under 365 nm UV excitation. (**B1**) Color coordinate value of the printed green QD-PDMS film. (**B2**) Enlarged color coordinate value of the printed green QD-PDMS film at different positions.

## Data Availability

The data presented in this study are available upon request from the corresponding author.

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
