# Peer review of "Development of Uniform Polydimethylsiloxane Arrays through Inkjet Printing"

_polymers, 2023, doi:10.3390/polym15020462_

Round 1
Reviewer 1 Report
Polymers-2134093
Development of Uniform polymer dots array through inkjet printing
Tu et al. proposed a polydimethylsiloxane (PDMS) based system to target the uniformities of inkjet printed polymer dots for polymer light emitting diode (polyLEDs) application. The authors studied the impact of solvent, viscosity, and concentration of PDMS on the qualities of printed dots array. Though the experiment was systematically designed and conducted. However, the current presentation demined the novelty of the study. The manuscript is more like a routine experimental report than a scientific paper. The author can rearrange the writing and try to make the work convincing.
Others:
1. What is the inkjet printing parameter?
2. PDMS should be reflected in the title.
3. What is the impact of the volume of the PDMS on the final diameter of the dots?
4. Does the substrate properties, such as surface roughness, hydrophobicity, etc., impact the size and shape of the dots?
5. Could it be possible to apply the PDMS dots for multi-color polyLEDs?
Reviewer 2 Report
This study has its merits because the authors performed the fabrication multicolor polymer light-emitting diode (polyLEDs), polymer base electronics, multicolour color conversion layers, and quantum dots light emitting diodes (QLEDs). However, the present version needs some improvements.
Major concerns:
* The Introduction section must be rewritten to be better understood by a broad audience.
* There is no clear hypothesis
* The Materials & Methods section is too short and incomplete.
* Results and Discussion section contains parts of the methodology.
* Results and Discussion section is too long with low amount of references,
Round 2
Reviewer 1 Report
Authors have addressed the queries pointed out by reviewers, hence, my recommendation for acceptance.